# Domains of the autism phenotype, cognitive control, and rumination as transdiagnostic predictors of DSM-5 suicide risk

Darren Hedley[1]*, Mirko Uljarević[2,3], Ru Ying Cai[4], Simon M. Bury[1], Mark A. Stokes[5], David W. Evans[6]

1 Olga Tennison Autism Research Centre, La Trobe University, Melbourne, Victoria, Australia, 2 University of Melbourne, Melbourne, Victoria, Australia, 3 Stanford Autism Center, Department of Psychiatry and Behavioral Sciences, School of Medicine, Stanford University, Stanford, California, United States of America, 4 Aspect Research Centre for Autism Practice, Melbourne, Victoria, Australia, 5 School of Psychology, Deakin University, Melbourne, Victoria, Australia, 6 Bucknell University, Lewisburg, Pennsylvania, United States of America

* D.Hedley@latrobe.edu.au

## Abstract

Suicide is a global health problem affecting both normative and clinical populations. Theoretical models that examine mechanisms underlying suicide risk across heterogeneous samples are needed. The present study explored core characteristics associated with autism spectrum disorder (ASD), a sub-population at high risk of suicide, as well as two dimensional cognitive constructs, as potential transdiagnostic predictors of suicidal ideation in a clinically diverse sample. Participants ($n = 1851$, 62% female) aged 18 to 89 years completed online questionnaires assessing: social communication difficulties; insistence on sameness; cognitive control; and rumination. Forty-three percent of participants reported the presence of at least one neurodevelopmental or neuropsychiatric disorder. One third of the sample reported some suicidal ideation (SI), and 40 percent met the threshold for concern for depression. All hypothesized constructs were associated with SI and depression and, with the exception of rumination, contributed significantly to SI. Participants reporting SI returned significantly higher social communication difficulties and insistence on sameness, and lower levels of cognitive control than those reporting no-SI. The study was limited by the use of a cross-sectional sample assessed with self-report measures. All diagnoses were self-reported and the study was additionally limited by the use of a single item indicator of suicidal ideation. These findings support a role for constructs associated with the ASD phenotype and associated broad cognitive domains as potential risk factors underlying suicidal ideation in a large clinically diverse sample. Our findings suggest directions for future longitudinal research studies, along with specific targets for suicide prevention and clinical practice.

**Data Availability Statement:** The data that support the findings of this study are openly available at "OSF" at https://doi.org/10.17605/OSF.IO/C2AP3. Additional correspondence or queries concerning

the data sample should be directed to David W.
Evans, PhD, Department of Psychology, Bucknell
University, Lewisburg, PA, 17837; e-mail:
dwevans@bucknell.edu.

**Funding:** Research reported in this study was
supported by Bucknell University Scholarly
Development Grant awarded to DWE, a Suicide
Prevention Australia National Suicide Prevention
Research fellowship awarded to DH, and a
Discovery Early Career Researcher Award from the
Australian Research Council awarded to MU. The
corresponding author had full access to all the data
in the study and final responsibility for the decision
to submit the report for publication. The content is
solely the responsibility of the authors and has not
been approved or endorsed by Suicide Prevention
Australia, Bucknell University, or the Australian
Research Council. The funder provided support in
the form of salaries for authors DH and MU and
research materials for DWE, but did not have any
additional role in the study design, data collection
and analysis, decision to publish, or preparation of
the manuscript. The specific roles of these authors
are articulated in the 'author contributions' section.

**Competing interests:** I have read the journal's
policy and the authors of this manuscript have the
following competing interests: DH is supported by
a Suicide Prevention Australia National Suicide
Prevention Research fellowship. MU is supported
by a Discovery Early Career Researcher Award
from the Australian Research Council
(DE180100632). DWE declares that he receives
funding from Roche Pharmaceutical for consulting
on the use of the Childhood Routines Inventory-
Revised (CRI-R). This does not alter our adherence
to PLOS ONE policies on sharing data and
materials.

## Introduction

Suicide and attempted suicide are major public health concerns, with suicide being a leading cause of death globally [1]. The World Health Organization estimates over 800,000 people die annually as a result of suicide, and it is the leading cause of death in youth aged 15 to 19 years [1]. Suicide is defined as the act of deliberately killing oneself, whereas suicide behavior refers to a range of behaviors including thinking about suicide (ideation), planning for suicide, non-fatal suicide attempt, as well as suicide [1]. To advance research in suicide prevention, O'Connor and Portzky, along with multiple international experts, identified key future developments and challenges in the field [2]. These included the need for more research into the testing and application of theoretical models of suicidal behavior, refining the understanding of sub-groups of people at risk in order to develop tailored interventions, and consideration of trans-diagnostic theoretical frameworks and models that better address the heterogeneity between people who experience suicidal behavior.

In the present study we adopt a dimensional approach, based on principles derived from the National Institute of Mental Health Research Domain Criteria (RDoC) [3, 4], as well as the support for a continuum between autistic traits in clinical cases and the general population [5, 6], to explore the contribution of autistic traits to heightened suicidal ideation in a large, clinically diverse sample. Autistic traits were selected due to the heightened risk of suicide amongst clinical cases (i.e., a sub-group at risk) [2], as well as recent evidence of an association between elevated autistic-related traits and global suicide risk [7].

Suicide risk has traditionally been studied within the context of single disorders (e.g., most notably, depression) [8]. However, suicide rates are elevated across a wide range of psychiatric disorders, and are compounded by the presence of comorbidities [9–11]. In addition, suicide risk is elevated in those with subclinical traits who may not meet criteria for a formal diagnosis of a psychiatric disorder [8, 11, 12]. Better understanding of sub-groups at risk, embedded within a transdiagnostic framework, is suggested to be a useful approach to improving understanding of mechanisms that underpin elevated suicide risk [2]. Notably, recent research has highlighted particularly high risk for suicidal behavior––including death by suicide––among individuals with autism spectrum disorder (ASD) who also exhibit increased levels of other neuropsychiatric symptoms [13–17]. Furthermore, autistic traits are present in other (i.e., non-ASD) clinical groups [18] and in the general population [5, 6, 19, 20]. Thus, traits present in clinical cases of ASD (e.g., social communication difficulties, cognitive rigidity and insistence on sameness) might also underlie psychiatric difficulties across normative and clinical samples [5, 6]; for example, ASD traits are found to be associated with a range of negative outcomes (e.g., anxiety; depression) [21, 22], including suicide [23]. Based on these observations, a prudent approach towards better understanding of the suicide risk and personalization of treatment is to adopt a dimensional and individual differences approach [24, 25].

Indeed, it is now widely understood that a range of domains transect categorical diagnostic boundaries, and these domains might provide predictive value in explaining phenomena over and beyond, and independent from, specific diagnoses [4, 24–26]. A dimensional approach–– which focuses on identifying specific risk and protective factors and attempts to understand underlying mechanisms––is therefore likely to provide a useful framework for understanding suicide risk and behavior across clinically diverse samples [24, 25, 27]. Recently, estimates of dimensional psychopathology derived from RDoC [3, 4] and applied to hospital discharge documentation were found to be associated with patient suicide and accidental death [27]; thereby demonstrating a potential application of dimensional frameworks to suicide prevention. However, to our knowledge, studies have yet to explore the nature of the interaction between constructs or traits associated with ASD, and transdiagnostic risk and resilience

factors such as cognitive control and positive and negative valence, in predicting suicidality in a large community sample spanning normative and atypical development. Furthermore, we currently lack insight into how elevated traits associated with ASD alone, and in combination with other neuropsychiatric symptoms, relate to suicidality in community samples.

### Autism phenotype as a risk factor for suicide

There is considerable evidence of heightened risk of suicidal behavior in people with ASD [13, 28–31], with suicide being the most significant predictor of premature mortality in individuals with ASD who do not have co-occurring intellectual disability (ID), as well as a significant risk factor in those with ID [14, 32]. Additionally, ASD trait severity is increased in adults with ASD who have planned or attempted suicide compared to those who do not have a lifetime history of planned or attempted suicide [13]. This suggests ASD trait severity may be a risk marker for suicide behavior in people with clinical ASD diagnoses. In terms of mechanisms, research with ASD clinical samples suggests ASD trait severity indirectly increases suicide risk through depression [30].

There is additional reason to believe that ASD traits may be an important risk factor for suicide behavior in broader clinical and non-clinical populations. From the standpoint of categorical diagnostic classifications, ASD commonly co-occurs with other neurodevelopmental and neuropsychiatric disorders [17], thereby compounding risk through the presence of multiple (vs. single) disorders. Traits or characteristics associated with ASD are normally distributed throughout the general population [33, 34] and, if viewed dimensionally, tend to be associated with elevated traits and symptoms of other disorders [35, 36]. There is emerging evidence that ASD traits are risk markers for suicide in people who do not have a diagnosis of ASD, including both non-clinical [7, 23, 28, 37, 38] and clinical (e.g., first episode psychosis) [17] populations.

Research with non-ASD populations suggests a direct relationship between ASD traits and suicidal behavior. In a non-clinical sample of young adults, Pelton and Cassidy [38] examined the relationship between ASD traits (broadly assessed with the Autism Spectrum Quotient, AQ) [33] and suicidal behavior within the context of the Interpersonal-Psychological Theory of Suicide (ITPS) [39]. ASD traits were found to significantly correlate with suicidal behavior, and this relationship was mediated by burdensomeness and thwarted belonging, suggesting a possible mechanism whereby social difficulties, which characterize ASD, may increase vulnerability to social risk factors for suicidal behavior. ASD traits also independently predicted variance in suicidal behavior in adults from the general population [28] and active military service members [23], supporting the hypothesis that heightened ASD traits increase risk for suicidal behavior in non-clinical populations. A study by Upthegrove et al. [17] examined the contribution of ASD traits to depression and suicide in a healthy, non-help seeking population, and in individuals experiencing first episode psychosis. Traits of ASD and psychosis were associated with increased levels of depressive symptoms in the non-help seeking population, and ASD traits and positive symptoms were associated with increased depressive symptoms, hopelessness, and suicidal ideation in the clinical sample. However, further research is required to assess the contribution of distinct constructs or characteristics directly associated with ASD, as well as broader but associated cognitive difficulties, to suicide risk.

Together, these studies suggest that traits associated with the ASD phenotype contribute to psychopathology. Moreover, it is plausible that ASD traits increase suicide risk either directly, or indirectly through depression and mediators such as hopelessness [17], loneliness, and low perceived social support [30], or burdensomeness and belonging [23, 38]. Furthermore, as has been demonstrated above, ASD traits may be present at the clinical or sub-clinical level thereby

affecting a larger sector of the population. To gain better understanding of the association between ASD traits and suicidal behavior, two important gaps in the research must be addressed. First, research is needed to tease apart those aspects of the ASD phenotype that confer risk for suicidality; for example, social communication difficulties and insistence on sameness or perseveration are factors that have been linked to suicidal behavior [24], and are also defining characteristics of ASD [40]. Second, it is important to understand how different aspects of the ASD phenotype interact with other transdiagnostic domains (e.g., cognitive control, negative valence) to predict suicidal behavior. This is clinically important as some of these factors might be modifiable through targeted treatment.

### Mechanisms underpinning suicide risk and the autistic phenotype

Diagnostically, ASD is characterized by persistent impairments in social communication and interaction, and the presence of restricted and repetitive patterns of behavior, interests and activities, inclusive of hyper- or hypo-reactivity to sensory stimuli [40]. Of these core symptoms, social communication difficulties and cognitive rigidity or insistence on sameness, that are also distributed across normative and clinical samples, may be particularly important risk factors for depression and suicidality [24, 25]. It is therefore important to understand how these factors individually, and collectively, interact to predict suicidal behavior within a clinically diverse, transdiagnostic sample.

Rumination and cognitive rigidity, which are associated with the restricted and repetitive domains of ASD [41], are not specific to ASD and are distributed across the general population, forming part of the RDoC Negative Valence system [3, 4]. Several studies have demonstrated a link between cognitive rigidity, which is associated with externalizing disorders [42], and suicide risk and behavior [42–44]. Rumination, on the other hand, is associated with internalizing symptoms [45] and depressive symptoms both in ASD [41, 45] and non-ASD populations [46]. Specifically, rumination has been found to predict the onset and duration of depression, and is associated with self-harm and suicidal ideation [47, 48].

In addition to the core symptoms, people with ASD often present with difficulties in broad cognitive domains including executive function and cognitive control [49–52], with these difficulties likely underpinning cognitive and behavioral rigidity, as well as social communication difficulties [53]. However, these deficits are not specific to ASD, but also feature across neurodevelopmental and neuropsychiatric disorders, as well as the general population, potentially leading to poor outcomes, which include suicide risk and behavior [54, 55].

The aims of the present study were to examine (1) the contribution of the two core clinical domains of ASD—social communication difficulties, insistence on sameness—on suicide risk (assessed using DSM-5 suicidal ideation; SI) and (2) the additional contribution and interaction of two key dimensional constructs––cognitive control and rumination. We predict that each of the identified constructs will independently contribute to SI, controlling for depression.

## Materials and methods

### Participants

Participants were 1851 (62.3% female) individuals aged 18–89 years ($M = 37.09$, $SD = 12.28$). The recruitment strategy followed that of previously published research and conducted recruitment online using Survey Sampling International (SSI; Shelton, CT) [56, 57], an online recruitment platform that specializes in recruiting demographically representative samples for scientific research in the United States and that is similar to other established and reliable commercial data recruitment platform (e.g., Prolific Academic, Amazon's Mechanical Turk)

[58–60]. Eligible participants were provided with a Qualtrics [61] link to the survey questionnaires. Participant demographics are provided in Table 1. The resultant sample was generally representative of the US population for race (although there were fewer Hispanics/Latinos in the study sample), income, education, and rural and urban populations [62], representing all 50 States as well as the District of Columbia (S1 Table). Given the sample consisted of a higher portion of females than males, demographic variables were explored between genders using Pearson's chi-squared test (Table 1). Race was proportionately distributed across gender, except for a slightly higher proportion of Hispanic males relative to females. Females were also more likely to report having more than one racial identity than males. Male and female participants differed significantly on highest education level achieved, household income, and marital status. Because the sample was otherwise representative of the general population, lifetime presence of neurodevelopmental and neuropsychiatric disorders (43.5%) was consistent with that reported for the United States (46.4%) [63]. Females reported a relatively higher number of psychiatric diagnoses than males overall, including significantly more diagnoses of anxiety, depression, and approaching significance for post-traumatic stress disorder; however, relatively more males than females reported a diagnosis of schizophrenia. Approximately one quarter of the sample reported taking medications for their condition, the difference in medication use between males and females was not statistically significant.

## Procedures

The research was approved by Bucknell University, Institutional Review Board (DWE's home institution). All participants reviewed an information document and were informed that participation was voluntary prior to agreeing to participate in the study. Online consent was received from all study participants.

**Construct items.**   To measure the constructs, specific items or subscales were selected from a series of measures after careful review by the first and second authors. The second and last author are additionally authors of one of the measures, the Adult Routines Inventory (ARI) [57]. All individual scale items were further reviewed to minimize the risk of introduced covariance between constructs. Items for each construct along with scoring information are provided in S1 Appendix.

*Social Communication Difficulties* were assessed using items specifically designed to evaluate these difficulties in those with ASD. These were drawn from the Autism Spectrum Quotient, an instrument designed to detect ASD traits in people with average or above intelligence quotient (IQ) [64]. Higher scores reflect greater social communication difficulties. These items were not used for the presence of ASD, but simply only social communication difficulties. Similarly, *Insistence on Sameness* was assessed with items drawn from a measure that evaluates this in part, the ARI [57]. The selected items assess routines, habits, and "compulsive-like" restricted and repetitive behaviors often seen in disorders such as OCD and ASD. Higher scores reflect greater rigidity. *Cognitive Control* was assessed with the Attentional and Inhibitory Control scales of the Adult Temperament Questionnaire [65]. Higher scores indicate greater control. *Rumination* was assessed with all 3-items from the Penn State Worry Questionnaire, ultra-brief version (PSWQ-3), which assesses pathological worry [66]. Higher scores indicate increased worry. *Depression* and *Suicidal Ideation* were assessed with three items from the adult version of the DSM-5 Level 1 Cross-Cutting (CC) Symptom Measure [67, 68], a self-rated measure of mental health domains that was developed by the DSM-5 Task Force and Work Groups [69]. Depressive symptoms are indicated by two items and suicide risk is assessed with a single item which assesses SI. Respondents are asked to consider how much or how often they have been bothered by a specific symptom during the last two weeks. A score

**Table 1. Demographics including neurodevelopmental and neuropsychiatric disorders, and gender comparisons.**

| Variable | Label | Male | Female | Total | General population data | Variable × Gender [95% BCa CI] |
|---|---|---|---|---|---|---|
| | | | n (%) | | | |
| n | | 698 (37.7%) | 1153 (62.3%) | 1851 (100%) | – | |
| | | | | | 2010 US Census, % | |
| Race | White | 510 (73.1%) | 821 (71.2%) | 1331 (71.9%) | 72.4% | $\chi^2(1) = 0.745$, $p = .388$ |
| | Black/African American | 73 (10.5%) | 129 (11.2%) | 202 (10.9%) | 12.6% | $\chi^2(1) = 0.238$, $p = .626$ |
| | Hispanic | 54 (7.7%) | 62 (5.4%) | 116 (6.3%) | 16.4% | $\chi^2(1) = 4.12$, $p = .042$ |
| | Asian | 34 (4.9%) | 44 (3.8%) | 78 (4.2%) | 4.8% | $\chi^2(1) = 1.20$, $p = .274$ |
| | Native Hawaiian/Pacific Islander | 2 (0.3%) | 4 (0.3%) | 6 (0.3%) | 0.2% | $\chi^2(1) = 0.049$, $p = .825$ |
| | Native American | 3 (0.4%) | 14 (1.2%) | 17 (0.9%) | 0.9% | $\chi^2(1) = 2.94$, $p = .086$ |
| | More than one | 21 (3%) | 72 (6.2%) | 93 (5%) | – | $\chi^2(1) = 9.54$, $p = .002$ |
| | Other[a] | 1 (0.1%) | 6 (0.5%) | 7 (0.4%) | – | $\chi^2(1) = 1.64$, $p = .200$ |
| Education | Less than high school | 16 (2.3%) | 31 (2.7%) | 47 (2.5%) | – | $\chi^2(7) = 125. 05$, $p < .001$[d] |
| | High school or GED | 169 (24.2%) | 316 (27.4%) | 485 (26.2%) | – | – |
| | Some college | 117 (16.8%) | 320 (27.8%) | 437 (23.6%) | – | – |
| | 2-year college degree | 67 (9.6%) | 182 (15.8%) | 249 (13.5%) | – | – |
| | 4-year college degree (BA, BS) | 207 (29.7%) | 250 (21.7%) | 457 (24.7%) | – | – |
| | Master's degree (MA, MS) | 72 (10.3%) | 40 (3.5%) | 112 (6.1%) | – | – |
| | Doctoral degree (PhD) | 23 (3.3%) | 4 (0.3%) | 27 (1.5%) | – | – |
| | Professional degree (MD, JD) | 23 (3.3%) | 8 (0.7%) | 31 (1.7%) | – | – |
| | Not reported | 4 (0.6%) | 2 (0.2%) | 6 (0.3%) | – | – |
| | | | | | 2014 Congressional, % | |
| Income | < $10,000 | 29 (4.2%) | 87 (7.5%) | 116 (6.3%) | 7.3% | $\chi^2(11) = 293.37$, $p < .001$[d] |
| | $10,000–$19,999 | 35 (5%) | 77 (6.7%) | 112 (6.1%) | 11.5% | – |
| | $20,000–$29,999 | 55 (7.9%) | 131 (11.4%) | 186 (10%) | 10.9% | – |
| | $30,000–$39,999 | 52 (7.4%) | 150 (13%) | 202 (10.9%) | 10% | – |
| | $40,000–$49,999 | 60 (8.6%) | 126 (10.9%) | 186 (10%) | 8.9% | – |
| | $50,000–$59,999 | 55 (7.9%) | 123 (10.7%) | 178 (9.6%) | 7.6% | – |
| | $60,000–$69,999 | 45 (6.4%) | 87 (7.5%) | 132 (7.1%) | 6.8% | – |
| | $70,000–$79,999 | 73 (10.5%) | 94 (8.2%) | 167 (9%) | 5.9% | – |
| | $80,000–$89,999 | 47 (6.7%) | 38 (3.3%) | 85 (4.6%) | 4.9% | – |
| | $90,000–$99,999 | 57 (8.2%) | 66 (5.7%) | 123 (6.6%) | 4% | – |
| | $100,000–$149,999 | 105 (15%) | 119 (10.3%) | 224 (12.1%) | 12.4% | – |
| | $\geq$ $150,000 | 76 (10.9%) | 44 (3.8%) | 120 (6.5%) | 9.5% | – |
| | Not reported | 9 (1.3%) | 11 (1%) | 20 (1.1%) | – | – |
| Marital status | Single, never married | 142 (20.3%) | 250 (21.7) | 392 (21.2%) | – | $\chi^2(4) = 40.85$, $p < .001$[d] |
| | Married | 522 (74.8%) | 752 (65.2%) | 1274 (68.8%) | – | – |
| | Separated | 7 (1%) | 30 (2.6%) | 37 (2%) | – | – |
| | Divorced | 24 (3.4%) | 87 (7.5%) | 111 (6%) | – | – |
| | Widowed | 1 (0.1%) | 32 (2.8%) | 33 (1.8%) | – | – |
| | Not reported | 2 (0.3%) | 2 (0.2%) | 4 (0.2%) | – | – |
| Diagnosis[b] | None | 435 (62.3%) | 610 (52.9%) | 1045 (56.5%) | – | $\chi^2(1) = 15.68$, $p = < .001$ |
| | Anxiety | 117 (16.8%) | 355 (30.8%) | 472 (25.5%) | – | $\chi^2(1) = 45.03$, $p = < .001$ |
| | Depression | 107 (15.3%) | 351 (30.4%) | 458 (24.7%) | – | $\chi^2(1) = 53.33$, $p = < .001$ |
| | ADD/ADHD | 44 (6.3%) | 82 (7.1%) | 126 (6.8%) | – | $\chi^2(1) = 0.448$, $p = .503$ |
| | Bipolar Disorder | 32 (4.6%) | 73 (6.3%) | 105 (5.7%) | – | $\chi^2(1) = 2.48$, $p = .115$ |
| | Obsessive Compulsive Disorder | 25 (3.6%) | 42 (3.6%) | 67 (3.6%) | – | $\chi^2(1) = 0.003$, $p = .960$ |
| | Autism Spectrum Disorder | 8 (1.1%) | 7 (0.6%) | 15 (0.8%) | – | $\chi^2(1) = 1.57$, $p = .210$ |

*(Continued)*

**Table 1.** (Continued)

| Variable | Label | Male | Female | Total | General population data | Variable × Gender [95% BCa CI] |
|---|---|---|---|---|---|---|
| | | | n (%) | | | |
| | Tic Disorder | 6 (0.9%) | 4 (0.3%) | 10 (0.5%) | – | $\chi^2(1) = 2.13$, $p = .145$ |
| | Post-Traumatic Stress Disorder | 1 (0.1%) | 9 (0.8%) | 10 (0.5%) | – | $\chi^2(1) = 3.29$, $p = .070$ |
| | Schizophrenia | 6 (0.9%) | 2 (0.2%) | 8 (0.4%) | – | $\chi^2(1) = 4.76$, $p = .029$ |
| | Personality Disorder | 1 (0.1%) | 2 (0.2%) | 3 (0.2%) | – | $\chi^2(1) = 0.024$, $p = .876$ |
| | Other[c] | 35 (5%) | 20 (1.7%) | 55 (3%) | – | $\chi^2(1) = 1.61$, $p = .205$ |
| Medication | | 158 (22.6%) | 299 (25.9%) | 457 (24.7%) | – | $\chi^2(1) = 2.57$, $p = .109$ |

[a]Hebrew Israelite, Indigenous, German (all $n = 1$), mixed ($n = 2$), not reported ($n = 2$).

[b]Sum of diagnoses is more than total number of individuals due to selecting multiple options.

[c]Other reported diagnoses were mostly non-psychiatric diagnoses and included anger/rage, arthritis, back/shoulder pain ($n = 2$), bronchitis, cancer (unspecified = 1, thyroid = 1), celiac disease, eczema, epilepsy ($n = 2$), diabetes ($n = 3$), gastroesophageal reflux disease (GERD), high blood pressure/cholesterol, human papillomavirus (HPV), insomnia, migraines ($n = 3$), Meniere's disease, menopause, multiple sclerosis, obesity, trichotillomania, panic disorder, not reported ($n = 28$).

[d]Group comparison statistics are reported for the overall category only.

$\geq 2$ on any item for depression and $\geq 1$ for SI serve as a clinical guide for additional inquiry and follow up.

**Data cleaning and analysis.** No more than 1% ($M = 0.303$, $SD = .19$, Range = 0–1%) of data were missing for any questionnaire item overall, and Little's MCAR test was not significant, $p = .895$ [70]. Thus, following Tabachnick and Fidell [71], cases with missing data on any of the questionnaires were deleted ($n = 77$, 3.8%). Where appropriate to do so, analyses were conducted using bootstrapping with 5000 resamples to provide more robust statistics, and 95% confidence intervals (BCa 95% CI) were used to interpret significance [71, 72]. Correlational analysis was used first to explore relationships between study variables. Bonferroni adjustment was used to account for multiple comparisons. Multiple linear regression was then run to identify factors contributing to suicidal ideation. Prior to performing the regression analysis the distribution of the residuals of the regression was reviewed for normality [73]. A Predicted Probability (P-P) plot was examined for normality with all constructs entered with suicidal ideation entered as the dependent variable. Examination of the P-P plot revealed that the residuals were normally distributed. All VIF values were below 10 (range 1.06–1.99).

Bootstrapped analysis of covariance (ANCOVA) controlling for age and depression were used to compare participants reporting no suicidal ideation (SI = 0) and those reporting presence of suicidal ideation (SI $\geq$ 1) on key study variables.

## Results

The data that support the findings of the study are openly available at "OSF" at https://doi.org/10.17605/OSF.IO/C2AP3 [74].

The DSM-5 CC Symptom Measure was examined first to determine risk for depression and SI (Table 2). Overall, approximately 42–44% of the total sample met the 'threshold for further inquiry' for depression, and 33% met the threshold for follow-up for suicide risk due to presence of SI [67]. Means and standard deviations, and correlation coefficients between study variables are provided in Table 3. Given study variables were significantly correlated with age, partial correlations controlling for age were also examined although the pattern of results was unaffected. Social Communication Difficulties were significantly correlated with all variables, with effect sizes ranging from small to medium ($r_p = .216–.365$). Study variables were all significantly correlated with Depression and SI in the expected directions, with effect sizes in the

**Table 2. Distribution of scores on DSM-5 CC Symptom Measure, depression and suicidal ideation (N = 1851).**

| | n (%) | | |
|---|---|---|---|
| Score | Depression item 1[a] | Depression item 2[b] | Suicidal ideation |
| None (0) | 574 (31%) | 594 (32.1%) | 1241 (67%) |
| Slight (1) | 463 (25%) | 478 (25.8%) | 207 (11.2%) |
| Mild (2) | 404 (21.8%) | 384 (20.7%) | 187 (10.1%) |
| Moderate (3) | 258 (13.9%) | 260 (14.0%) | 143 (7.7%) |
| Severe (4) | 152 (8.2%) | 135 (7.3%) | 73 (3.9%) |
| Threshold for further inquiry[c] | 814 (44%) | 779 (42.1%) | 610 (33%) |

[a]"Little interest or pleasure in doing things".

[b]"Feeling down, depressed, or hopeless".

[c]Depression score $\geq$ 2, Suicidal ideation $\geq$ 1.

small to large range ($r_p$ = -.117–.590). Effect sizes for Social Communication Difficulties were in the medium range for Depression and SI and, as expected, SI was strongly correlated with Depression. In terms of the other variables, Insistence on Sameness, Rumination (both positively) and Attentional Control (negatively) were most strongly associated with Depression, and Attentional Control (negatively) was most strongly associated with SI. Thus, all of the hypothesized constructs were found to be significantly associated with SI thereby warranting their inclusion in the linear regression analysis.

## Regression analysis

Table 4 presents the results of the linear regression model predicting SI. All hypothesized constructs were included in the model. Age was controlled for by including it in the model. The full model accounted for 43.3% of variance in SI scores, $F(7, 1843)$ = 201.19, $p < .001$. Social Communication Difficulties significantly predicted SI, with the b-weight revealing that for each unit increase in Social Communication Difficulties, SI increased by 0.085 units. Similarly,

**Table 3. Study variables (M, SD, range, normality) with Pearson's bootstrapped correlations (upper panel, shaded), and partial correlations (lower panel) controlling for age (n = 1851).**

| Variable | M | SD | Range | Shapiro-Wilk | 2. | 3. | 4. | 5. | 6. | 7. | 8. |
|---|---|---|---|---|---|---|---|---|---|---|---|
| 1. Age (years) | 37.09 | 12.28 | 18–89 | .912* | -.094* [-.14,-.05] | -.132* [-.18,-.09] | .173* [.13,.22] | .169* [.12,.22] | -.139* [-.18,-.09] | -.181* [-.22,-.14] | -.199* [-.24,-.15] |
| 2. Social Communication Difficulties | 9.96 | 2.61 | 5–20 | .975* | – | .115* [.07,.16] | -.362* [-.40,-.32] | -.280* [-.33,-.23] | .152* [.11,.20] | .206* [.17,.25] | .318* [.28,.36] |
| 3. Insistence on Sameness | 44.30 | 14.08 | 15–75 | .989* | .257* [.21,.30] | – | -.432* [-.47,-.39] | -.178* [-.22,-.13] | .487* [.45,.53] | .428* [.39,.47] | .386* [.35,.43] |
| 4. Attentional Control | 22.62 | 6.60 | 5–35 | .983* | -.364* [-.40,-.32] | -.419* [-.46,-.38] | – | .384* [.34,.42] | -.587* [-.62,-.55] | -.537* [-.57,-.50] | -.436* [-.47,-.40] |
| 5. Inhibitory Control | 30.40 | 6.17 | 11–49 | .974* | -.292* [-.34,-.25] | -.159* [-.20,-.11] | .365* [.32,.41] | – | -.270* [-.31,-.23] | -.227* [-.27,-.18] | -.146* [-.18,-.11] |
| 6. Rumination | 9.28 | 4.79 | 3–18 | .929* | .216* [.17,.26] | .478* [.44,.52] | -.578* [-.61,-.54] | -.254* [-.30,-.21] | – | .600* [.57,.63] | .388* [.35,.43] |
| 7. Depression | 2.82 | 2.36 | 0–8 | .914* | .276* [.23,.32] | .414* [.37,.46] | -.523* [-.56,-.49] | -.202* [-.25,-.16] | .590* [.56,.62] | – | .602* [.57,.64] |
| 8. Suicidal ideation | .703 | 1.16 | 0–4 | .652* | .365* [.33,.40] | .371* [.33,.41] | -.416* [-.45,-.38] | -.117* [-.15,-.08] | .371* [.33,.41] | .588* [.55,.62] | – |

*$p < .001$.

**Table 4. Linear regression model of predictors of suicidal ideation.**

|  | b | SEB[a] | β | p-value | BCa 95% CI |
|---|---|---|---|---|---|
| **Constant** | -1.076 | 0.224 | – | < .001 | **-1.508, -.617** |
| Age | -0.007 | 0.002 | -0.079 | .001 | **-0.011, -0.004** |
| Social Communication Difficulties | 0.085 | 0.009 | 0.192 | < .001 | **0.067, 0.104** |
| Insistence on Sameness | 0.012 | 0.002 | 0.144 | < .001 | **0.009, 0.015** |
| Attentional Control | -0.015 | 0.004 | -0.085 | .001 | **-0.023, -0.006** |
| Inhibitory Control | 0.015 | 0.004 | 0.077 | < .001 | **0.008, 0.021** |
| Rumination | -0.010 | 0.006 | -0.042 | .086 | -0.022, 0.001 |
| Depression | 0.238 | 0.011 | 0.484 | < .001 | **0.213, 0.264** |

$R^2$ = .433, $F(7, 1843)$ = 201.19, $p < .001$. BCa 95% confidence intervals that do not cross zero are bolded.

[a]SEB: the standard error for the unstandardized beta.

95% bias corrected and accelerated confidence intervals and standard errors based on 5000 bootstrap samples.

Insistence on Sameness was also identified as a significant predictor of SI, with the b-weight revealing that for each unit increase in Insistence on Sameness, SI increased by 0.012 units. Attentional and Inhibitory Control both significantly predicted SI, with the b-weights revealing that for each unit increase in Attentional Control, SI decreased by 0.015 units, and for each unit increase in Inhibitory Control, SI increased by 0.015 units. Rumination was not a significant predictor of SI when entered in the model with the other variables, with each unit increase in Rumination associated with a decrease in SI of -0.010 units. Overall, Depression made the largest contribution to SI (β = 0.484). Comparing Social Communication to Insistence on Sameness; Social Communication Difficulties (β = 0.192) was relatively more important than Insistence on Sameness (β = 0.144). These two core variables shared some variance, but correlations in Table 3 reveal that these were largely independent contributions. Attentional Control (β = -0.085) and Inhibitory Control (β = 0.077) made similar, yet relatively smaller contributions to the model.

## Suicidal ideation present versus not present comparisons

The sample was split into those reporting no suicidal ideation (SI = 0, n = 1241) and those reporting at least some ideation (SI ≥ 1, n = 610). Groups were compared on age, depression,

**Table 5. Means (SD) and bootstrapped ANCOVA comparisons between no-SI and SI groups on key variables controlling for age and depression.**

|  | No-SI (n = 1241) | | SI (n = 610) | | F-statistic | | | |
|---|---|---|---|---|---|---|---|---|
| Variable[a] | M | SD | M | SD | df = 1, 1847 | p-value[b] | BCa 95% CI[c] | Cohen's d [95% CI] |
| Social communication difficulties | 9.36 | 2.60 | 11.17 | 2.20 | 128.82 | < .001 | **-1.99, -1.39** | -0.73 [-0.88, -0.56] |
| Insistence on sameness | 41.16 | 13.61 | 50.70 | 12.80 | 17.51 | < .001 | **-4.66, -1.68** | -0.72 [-1.47, 0.30] |
| Attentional control | 24.51 | 6.30 | 18.79 | 5.45 | 36.53 | < .001 | **1.34, 2.67** | 0.95 [0.60, 1.38] |
| Inhibitory control | 31.23 | 6.76 | 28.71 | 4.29 | 7.62 | .003 | **0.354, 1.67** | 0.42 [0.04, 0.76] |
| Rumination | 8.07 | 4.55 | 11.72 | 4.30 | 1.07 | .316 | -0.714, 0.236 | -0.82 [-1.07, -0.48] |
| Covariates |  |  |  |  | df = 1, 1849 |  |  |  |
| Age | 38.82 | 12.20 | 33.56 | 11.67 | 78.06 | < .001 | **4.05, 6.44** | 0.44 [-0.24, 1.36] |
| Depression | 1.89 | 1.93 | 4.72 | 1.99 | 864.60 | < .001 | **-3.03, -2.63** | -1.45 [-1.56, -1.29] |

[a]Age and depression entered as covariates in the model.

[b]5000 samples bootstrapped p-value.

[c]BCa 95% confidence intervals that do not cross zero are bolded.

and the main study variables. Results of these analyses are presented in Table 5. Groups differed significantly on age, with those reporting no SI being overall older than those reporting presence of SI. Cohen's *d* effect size for the difference was in the small to moderate range. As would be expected, depression scores were also significantly higher in those reporting SI than the no SI group, with the difference returning a large effect size. Subsequently, bootstrapped ANCOVAs were used to compare the two groups on each of the main study variables, controlling for age and depression. Group membership did not have a significant effect on Rumination after controlling for age and depression in the model. There was a significant effect of group membership on core ASD related traits (i.e., Social Communication Difficulties, Insistence on Sameness) and cognitive variables (i.e., Attentional and Inhibitory Control). Thus, participants who reported some SI reported significantly greater Social Communication Difficulties, higher levels of Insistence on Sameness, and lower levels of Attentional and Inhibitory Control, than participants who did not report any SI.

## Discussion

The present study aimed to examine the contribution of social communication difficulties and insistence on sameness, representative of core features of ASD, as well as cognitive control and ruminative thinking, to DSM-5 suicidal ideation [67, 68] in a large online recruited sample comprising normative and clinically diverse individuals. Cognitive control is a potential trans-diagnostic risk factor for suicidal behavior that remains underexplored [24, 25], and is affected in ASD [49–51, 53, 75]. Similarly, rumination and cognitive rigidity/insistence on sameness have been shown to be associated with depression [46, 76] and suicidal ideation [47, 48], as well as ASD traits [45]. Social communication difficulties, which are associated with depression in non-ASD samples [77, 78], are relatively unexplored in terms of their contribution to suicide risk, but are core characteristics of the ASD phenotype. We were specifically interested to know whether each of these constructs provided a unique contribution to suicidal ideation after controlling for depressive symptoms.

One third of the sample met the DSM-5 CC threshold for further inquiry for suicide risk due to presence of suicidal ideation, with around 40 percent meeting the threshold for concern for depression. Correlational analyses revealed that higher scores on social communication difficulties, insistence on sameness and rumination, and lower scores on attentional and inhibitory control were all significantly associated with DSM-5 CC depression and suicidal ideation. Regression analysis controlling for depression revealed that all factors excepting rumination contributed significantly to DSM-5 CC suicidal ideation, with the full model accounting for 43 percent of variance in suicidal ideation scores. Comparison of participants reporting at least some versus no suicidal ideation, controlling for age and depression, revealed significantly higher levels of social communication difficulties and insistence of sameness, and lower levels of attentional and inhibitory control in the group reporting some suicidal ideation.

Our findings suggest a role for core ASD related traits in suicidal ideation, consistent with studies reporting a high level of risk in ASD clinical [13, 14, 28, 30] and non-ASD [15, 17, 23, 38] samples. Indeed, our findings indicate that social communication difficulties and insistence on sameness independently predicted suicidal ideation when controlling for cognitive risk factors and depression. Moreover, results of this study contribute to an emerging evidence base positing ASD traits as an important dimensional construct underlying suicide risk that cuts across diagnostic boundaries. Our study extends previous research [13, 17, 28, 30, 38] by deconstructing specific components (i.e., social communication difficulties, insistence on sameness) of the ASD phenotype that represent the two primary clinical domains of the disorder. Importantly, the identification of the role of domains associated with the ASD phenotype,

representative of a clinical group at high risk of suicide, contributes to understanding and development of a transdiagnostic and dimensional framework for understanding suicide risk [8, 24, 25, 27]. The development of such a model pinpoints specific targets for intervention, and strengthens the call for assessing both individuals with clinical diagnoses of ASD, and those with high levels of ASD traits, for suicide risk [17].

Additional work is needed, however, to further clarify the processes (e.g., social, cognitive) whereby ASD traits contribute to suicide risk. Previous research has identified social relationships and loneliness as potentially important to depression and suicide in individuals with ASD [30, 79, 80], the present study extends this model by also examining cognitive control and negative valence domains.

Our findings provide support for the contribution of poor cognitive control, cognitive rigidity, and ruminative thinking style to suicidal ideation. However, it is interesting to note that the association between rumination and suicidal ideation was no longer significant when controlling for depression, suggesting overlap between these two factors. While depression is highly prevalent in people with ASD [81, 82], the mechanisms underlying increased suicide risk in this population may represent an interaction between traits associated with the core characteristics of the diagnosis, associated cognitive dysfunction, and co-occurring psychiatric conditions. Moreover, our findings suggest these mechanisms are not limited to clinical cases, but may constitute transdiagnostic risk factors. Together, our findings are significant in that they a) represent an attempt at unpacking the mechanisms associated with the ASD phenotype that might contribute to increased suicide risk and b) contribute to the growing literature concerning dimensional, transdiagnostic risk factors and mechanisms underlying suicide risk and behavior.

## Strengths and limitations

This study was strengthened by our use of a large sample and theoretically informed constructs known to be associated with suicide risk. Our inclusion of constructs related to ASD traits, specifically social communication difficulties, RRBIs, and associated cognitive challenges represents a novel contribution to the literature. Nonetheless, the cross-sectional nature of the study limits our ability to infer causality. Findings are also limited due to reliance on self-report measures, including psychiatric diagnoses, and use of an online survey for data collection. Although our findings were generally consistent with the literature and theoretical background, it will be important to replicate our findings taking into consideration these methodological limitations. Future research will benefit from administration of cognitive assessments to better elucidate the effect of cognitive processes on suicide risk and more comprehensive, questionnaire and performance-based quantitative measures designed to capture strengths and weaknesses across different domains of social functioning, such as the SEL web [83] and the Stanford Social Dimensions Scale [84]. Suicide risk was assessed using the DSM-5 suicidal ideation screener incorporated into the cross-cutting symptom measure [67, 85], which was selected as it is relatively straightforward to administer using large scale online methodology, and because the presence of suicidal ideation has been shown to increase the likelihood of a suicide attempt [86]. However, the use of a single indicator or suicide risk limits our findings [87, 88]. Future research would benefit from a more comprehensive assessment of suicide risk and behavior.

## Future directions

Executive dysfunction, including cognitive rigidity and poor decision-making, may be a trait vulnerability for suicide risk [89]. Our findings concerning rumination, depression, and

suicidal ideation suggests an inter-relationship among these three constructs. Response Styles Theory [90] posits that rumination is a cognitive response to depressed mood. Our findings indicate that, although associated with depression and suicidal ideation, rumination itself is not predictive of suicidal ideation scores. Future research is required to disentangle the associations among these factors, and to understand better the potential contribution of repetitive thinking to suicide risk and behavior. Ultimately, this calls for the development of including genetic and biological, cognitive, and social elements (i.e., bio-psycho-social model) underlying suicide risk. The first step in this process is to identify those individual mechanisms and to demonstrate their links to suicide behavior.

## Conclusions

Despite the aforementioned limitations, our study demonstrated the potential role of ASD traits, particularly social communication difficulties and cognitive rigidity/insistence on sameness, and difficulties in broad cognitive domains associated with the ASD phenotype, as potential transdiagnostic factors underlying suicide risk. Our approach represents a shift away from disorder-specific research in an attempt at uncovering common mechanisms and risk factors for suicide behavior in individuals with no psychiatric diagnosis, individuals with diagnoses of one or more common psychiatric disorders, and individuals who may not have a formal diagnosis but who present with subthreshold symptomatology. These findings provide a roadmap for further longitudinal research and identify potential targets for intervention and clinical practice.

## Supporting information

**S1 Appendix. Scale items and ratings.** Items for each of the constructs used in the study. (DOCX)

**S1 Table. Participant distribution.** Distribution of participants between US States and the District of Columbia. (DOCX)

## Acknowledgments

We thank the individuals who participated in this study. DWE and MU designed the survey and collected the data. DH completed the literature review. DH and MU conceived of the report and prepared the tables. DH completed the data analysis and interpretation. DH and MU wrote the report, incorporating comments from all authors. MAS and DWE critically revised the report. All authors reviewed and approved the final submitted version.

## Author Contributions

**Conceptualization:** Darren Hedley, Mirko Uljarević.

**Data curation:** David W. Evans.

**Formal analysis:** Darren Hedley.

**Funding acquisition:** Darren Hedley, David W. Evans.

**Investigation:** Darren Hedley, David W. Evans.

**Methodology:** Darren Hedley, Mirko Uljarević, David W. Evans.

**Project administration:** David W. Evans.

**Writing – original draft:** Darren Hedley.

**Writing – review & editing:** Darren Hedley, Mirko Uljarević, Ru Ying Cai, Simon M. Bury, Mark A. Stokes, David W. Evans.

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
