## [Decision Letter · Decision Letter 0]

26 Oct 2020

PONE-D-20-29676

Domains of the autism phenotype, cognitive control, and rumination as transdiagnostic predictors of DSM-5 suicide risk

PLOS ONE

Dear Dr. Hedley,

Thank you for submitting your manuscript to PLOS ONE. After careful consideration, we feel that it has merit but does not fully meet PLOS ONE’s publication criteria as it currently stands. Therefore, we invite you to submit a revised version of the manuscript that addresses the points raised during the review process.

We look forward to receiving your revised manuscript.

Kind regards,

Vincenzo De Luca

Academic Editor

PLOS ONE

Journal Requirements:

2.Please provide additional details regarding participant consent. In the ethics statement in the Methods and online submission information, please ensure that you have specified (1) whether consent was informed and (2) what type you obtained (for instance, written or verbal, and if verbal, how it was documented and witnessed). If your study included minors, state whether you obtained consent from parents or guardians. If the need for consent was waived by the ethics committee, please include this information.

3.We note that you have indicated that data from this study are available upon request. PLOS only allows data to be available upon request if there are legal or ethical restrictions on sharing data publicly. For more information on unacceptable data access restrictions, please see http://journals.plos.org/plosone/s/data-availability#loc-unacceptable-data-access-restrictions.

4.Thank you for stating the following in the Competing Interests section:

[I have read the journal's policy and the authors of this manuscript have the following competing interests: DH is supported by a Suicide Prevention Australia National Suicide Prevention Research fellowship. MU is supported by a Discovery Early Career Researcher Award from the Australian Research Council (DE180100632). DWE declares that he receives funding from Roche Pharmaceutical for consulting on the use of the Childhood Routines Inventory-Revised (CRI-R). ].

5.Thank you for stating the following financial disclosure:

 [Research reported in this study was supported by Bucknell University Scholarly Development Grant awarded to DWE, a Suicide Prevention Australia National Suicide Prevention Research fellowship awarded to DH, and a Discovery Early Career Researcher Award from the Australian Research Council awarded to MU. The corresponding author had full access to all the data in the study and final responsibility for the decision to submit the report for publication. The content is solely the responsibility of the authors and has not been approved or endorsed by Suicide Prevention Australia, Bucknell University, or the Australian Research Council. The funders had no role in study design, data collection and analysis, decision to publish, or preparation of the manuscript. ].

We note that one or more of the authors is affiliated with the funding organization, indicating the funder may have had some role in the design, data collection, analysis or preparation of your manuscript for publication; in other words, the funder played an indirect role through the participation of the co-authors. If the funding organization did not play a role in the study design, data collection and analysis, decision to publish, or preparation of the manuscript and only provided financial support in the form of authors' salaries and/or research materials, please do the following:

Review your statements relating to the author contributions, and ensure you have specifically and accurately indicated the role(s) that these authors had in your study. These amendments should be made in the online form.

Confirm in your cover letter that you agree with the following statement, and we will change the online submission form on your behalf:

Reviewers' comments:

Reviewer's Responses to Questions

**Comments to the Author**

1. Is the manuscript technically sound, and do the data support the conclusions?

Reviewer #1: Partly

2. Has the statistical analysis been performed appropriately and rigorously? 

Reviewer #1: No

3. Have the authors made all data underlying the findings in their manuscript fully available?

Reviewer #1: Yes

4. Is the manuscript presented in an intelligible fashion and written in standard English?

Reviewer #1: Yes

5. Review Comments to the Author

Reviewer #1: This paper examines the relationship between autistic traits and suicidal ideation. I think the topic of this research is very important, and the results of this research can help us better understand both autism and suicidal thoughts. In addition, the study had a very large sample size (n=1851) with participants from a diverse background. Such large representative data can give us accurate estimates of the population parameters of interest; thus, the authors should definitely find a channel to publish these data. However, there are several major problems with the paper that need to be addressed:

1. The data analyses of the paper are not well done. The authors claim that they did hierarchical regression analyses but based on the description of the analysis procedure, they actually did stepwise regression (see page 20). Stepwise regression is usually used in exploratory studies where researchers try to search for important predictors for the criterion variable. However, this paper is clearly a confirmatory study. I don’t see any justification for using the stepwise regression. I recommend the authors just use regular multiple regression analyses.

2. The results section of the paper is very poorly written. The results section merely introduced the tables for the results; it did not describe the patterns of the results at all. In the next revision, please provide a summary of the results of each table and describe the results patterns in words (i.e., do not just present the results in tables).

3. The authors included a lot of tables (e.g., Tables 1, 2 and 3) that are not essential for addressing the research goal. I recommend the authors put these tables in the supplementary materials.

In addition, there are few minor problems/mistakes in the paper:

• On page 6, there is a sentence that says “ASD traits were found to significantly correlate with suicidal behaviour, and the relationship was mediated by burdensomeness and thwarted belonging….. ” Does it mean burdensomeness and thwarted belonging are two of the underlying mechanisms explaining the relationship between autism and suicidal thoughts? If yes, then why this is not addressed in the next section of the paper, which is about “mechanisms underspinning suicide risk and the autistic phenotype”?

• There is a typo on page 8 under the “Participants” section. It says “…. condiucted recruitment online using…. ” “Condiucted” is a typo.

• On page 15, please provide a citation for Little’s MCAR test. Also on page 15, I am confused why the authors found out that the data are not MCAR but still used listwise deletion.

• On page 8, the sentence, “The aims of the present study were to examine (1) the contribution of two key clinical domains––social communication difficulties, insistence on sameness––usually considered core features of ASD, but also present across a range of other disorders, that have been associated with suicidal risk and behavior, thereby deconstructing the impact of specific ASD domains on suicide risk (assessed using DSM-5 suicidal ideation; SI)….”, is very confusing to read. Please simplify the sentence.

6. PLOS authors have the option to publish the peer review history of their article (what does this mean?). If published, this will include your full peer review and any attached files.

Reviewer #1: No

---

## [Author Response · Author response to Decision Letter 0]

28 Oct 2020

Dear editor,

Thank you for the opportunity to revise and re-submit our manuscript. Please find below a detailed response to all comments. We thank the reviewer and the editor for their suggestion leading to what we believe is an improved manuscript. Of note, we have revised the Results section and analyses as per the suggestions of the Review, and appreciate the suggestions.

Response: We have carefully checked the manuscript against the PLOS ONE style templates and have revised the manuscript accordingly, including the reference list, title page, and table notes. 

2.Please provide additional details regarding participant consent. In the ethics statement in the Methods and online submission information, please ensure that you have specified (1) whether consent was informed and (2) what type you obtained (for instance, written or verbal, and if verbal, how it was documented and witnessed). If your study included minors, state whether you obtained consent from parents or guardians. If the need for consent was waived by the ethics committee, please include this information. If you are reporting a retrospective study of medical records or archived samples, please ensure that you have discussed whether all data were fully anonymized before you accessed them and/or whether the IRB or ethics committee waived the requirement for informed consent. If patients provided informed written consent to have data from their medical records used in research, please include this information.

Response: The study was online. Participants read an information statement about the study and provided online consent. These additional details have been added to the Method section and the Ethics Statement in editorialmanager, and the Cover Letter. The consent process now reads: 

“All participants reviewed an information document and were informed that participation was voluntary prior to agreeing to participate in the study. Online consent was received from all study participants.”

3.We note that you have indicated that data from this study are available upon request. PLOS only allows data to be available upon request if there are legal or ethical restrictions on sharing data publicly. For more information on unacceptable data access restrictions, please see http://journals.plos.org/plosone/s/data-availability#loc-unacceptable-data-access-restrictions.

 a) If there are ethical or legal restrictions on sharing a de-identified data set, please explain them in detail (e.g., data contain potentially sensitive information, data are owned by a third-party organization, etc.) and who has imposed them (e.g., an ethics committee). Please also provide contact information for a data access committee, ethics committee, or other institutional body to which data requests may be sent. b) If there are no restrictions, please upload the minimal anonymized data set necessary to replicate your study findings as either Supporting Information files or to a stable, public repository and provide us with the relevant URLs, DOIs, or accession numbers. For a list of acceptable repositories, please see http://journals.plos.org/plosone/s/data-availability#loc-recommended-repositories. We will update your Data Availability statement on your behalf to reflect the information you provide.

Response: All data are fully available without restrictions. Data are freely downloadable from Figshare, https://doi.org/10.26181/5e992fc659d7c. The reference for the dataset is: 

“Hedley D, Uljarević M, Cai RY, Bury SM, Stokes MA, Evans DW. Transdiagnostic predictors of dsm-5 suicide risk [raw dataset, Version 2]. Figshare. 2020. https://doi.org/10.26181/5e992fc659d7c”

The statement in editorialmanager has been updated to indicate that data are openly available and that additional queries concerning the data sample can be directed to author DWE.

“The data that support the findings of this study are openly available in “figshare” at https://doi.org/10.26181/5e992fc659d7c [89]. Additional correspondence or queries concerning the data sample should be directed to David W. Evans, PhD, Department of Psychology, Bucknell University, Lewisburg, PA, 17837; e-mail: dwevans@bucknell.edu.”

4.Thank you for stating the following in the Competing Interests section:

[I have read the journal's policy and the authors of this manuscript have the following competing interests: DH is supported by a Suicide Prevention Australia National Suicide Prevention Research fellowship. MU is supported by a Discovery Early Career Researcher Award from the Australian Research Council (DE180100632). DWE declares that he receives funding from Roche Pharmaceutical for consulting on the use of the Childhood Routines Inventory-Revised (CRI-R). ].

Please confirm that this does not alter your adherence to all PLOS ONE policies on sharing data and materials, by including the following statement: "This does not alter our adherence to PLOS ONE policies on sharing data and materials.” (as detailed online in our guide for authors https://protect-au.mimecast.com/s/dHriC4QZgRFApQopfjbrgz?domain=journals.plos.org). If there are restrictions on sharing of data and/or materials, please state these. Please note that we cannot proceed with consideration of your article until this information has been declared.

Please know it is PLOS ONE policy for corresponding authors to declare, on behalf of all authors, all potential competing interests for the purposes of transparency. PLOS defines a competing interest as anything that interferes with, or could reasonably be perceived as interfering with, the full and objective presentation, peer review, editorial decision-making, or publication of research or non-research articles submitted to one of the journals. Competing interests can be financial or non-financial, professional, or personal. Competing interests can arise in relationship to an organization or another person. Please follow this link to our website for more details on competing interests: https://protect-au.mimecast.com/s/dHriC4QZgRFApQopfjbrgz?domain=journals.plos.org

Response: The Competing Interests statement has been update to include the additional information. It now reads:

“I have read the journal's policy and the authors of this manuscript have the following competing interests: DH is supported by a Suicide Prevention Australia National Suicide Prevention Research fellowship. MU is supported by a Discovery Early Career Researcher Award from the Australian Research Council (DE180100632). DWE declares that he receives funding from Roche Pharmaceutical for consulting on the use of the Childhood Routines Inventory-Revised (CRI-R). This does not alter our adherence to PLOS ONE policies on sharing data and materials.”

5.Thank you for stating the following financial disclosure:

 [Research reported in this study was supported by Bucknell University Scholarly Development Grant awarded to DWE, a Suicide Prevention Australia National Suicide Prevention Research fellowship awarded to DH, and a Discovery Early Career Researcher Award from the Australian Research Council awarded to MU. The corresponding author had full access to all the data in the study and final responsibility for the decision to submit the report for publication. The content is solely the responsibility of the authors and has not been approved or endorsed by Suicide Prevention Australia, Bucknell University, or the Australian Research Council. The funders had no role in study design, data collection and analysis, decision to publish, or preparation of the manuscript. ].

We note that one or more of the authors is affiliated with the funding organization, indicating the funder may have had some role in the design, data collection, analysis or preparation of your manuscript for publication; in other words, the funder played an indirect role through the participation of the co-authors. If the funding organization did not play a role in the study design, data collection and analysis, decision to publish, or preparation of the manuscript and only provided financial support in the form of authors' salaries and/or research materials, please do the following:

1. Review your statements relating to the author contributions, and ensure you have specifically and accurately indicated the role(s) that these authors had in your study. These amendments should be made in the online form.

Response: The information relating to author contributions and roles on the online form are updated and correct.

2. Confirm in your cover letter that you agree with the following statement, and we will change the online submission form on your behalf:

Response: We confirm that funding organization did not play a role in the study design, data collection and analysis, decision to publish, or preparation of the manuscript and only provided financial support in the form of authors' salaries and/or research materials. We confirm that we agree with the statement in [2]. The following statement has been added to the Cover Letter:

“The funder provided support in the form of salaries for authors DH and MU and research materials for DWE, but did not have any additional role in the study design, data collection and analysis, decision to publish, or preparation of the manuscript. The specific roles of these authors are articulated in the ‘author contributions’ section.”

Reviewer comments

1. The data analyses of the paper are not well done. The authors claim that they did hierarchical regression analyses but based on the description of the analysis procedure, they actually did stepwise regression (see page 20). Stepwise regression is usually used in exploratory studies where researchers try to search for important predictors for the criterion variable. However, this paper is clearly a confirmatory study. I don’t see any justification for using the stepwise regression. I recommend the authors just use regular multiple regression analyses.

Response: The data analyses have been revised as suggested by the Reviewer. We now report the results of a regular multiple linear regression analysis. 

2. The results section of the paper is very poorly written. The results section merely introduced the tables for the results; it did not describe the patterns of the results at all. In the next revision, please provide a summary of the results of each table and describe the results patterns in words (i.e., do not just present the results in tables).

Response: We have followed APA guidelines (www.apastyle.apa.org) for reporting the various analyses, and do summarize the results/findings for each of the main analyses. In line with the Reviewer’s recommendation, we have reviewed and revised Results reporting adding summary information where we felt it was appropriate to do so. We are happy to provide further information if the Review can please clearly identify what/how they would like the Results section to be changed. Examples of summaries for each of the main tables/analyses are provided below: 

Table 2: “Overall, approximately 42–44% of the total sample met the ‘threshold for further inquiry’ for depression, and 33% met the threshold for follow-up for suicide risk due to presence of SI”.

Table 3: “Social communication difficulties were significantly correlated with all variables, with effect sizes ranging from small to medium (rp = .216–.365). Study variables were all significantly correlated with depression and SI in the expected directions, with effect sizes in the small to large range (rp = -.117–.590). Effect sizes for social communication difficulties were in the medium range for depression and SI and, as expected, SI was strongly correlated with depression. In terms of the other variables, insistence on sameness, rumination (both positively) and attentional control (negatively) were most strongly associated with depression, and attentional control (negatively) was most strongly associated with SI. Thus, all of the hypothesized constructs were found to be significantly associated with SI thereby warranting their inclusion in the linear regression analysis. ” 

Table 4: “Table 4 presents the results of the linear regression model predicting SI. All hypothesized constructs were included in the model. Age was controlled for by including it in the model. The full model accounted for 43.3% of variance in SI scores, F(7, 1843) = 201.19, p < .001. In addition to depression (t = 20.81), social communication difficulties (t = 9.99), insistence on sameness (t = 6.95), attentional (t = -3.44) and inhibitory (t = 3.99) control were all identified as significant predictors of SI; however, rumination (t = -1.70) was not found to be a significant predictor of SI when entered in the model with the other variables.”

Table 5: “Thus, participants who reported some SI reported significantly greater social communication difficulties, higher levels of insistence on sameness, and lower levels of attentional and inhibitory control, than participants who did not report any SI.” 

3. The authors included a lot of tables (e.g., Tables 1, 2 and 3) that are not essential for addressing the research goal. I recommend the authors put these tables in the supplementary materials.

Response: Table 1 presents sample demographics and is not presented in the results section, Table 2 summarizes clinical depression and suicidal ideation symptoms on the main symptom measure, and Table 3 presents means and correlations between the main study variables. We respectfully disagree with the Reviewer that these tables are better placed as Supplementary materials (which are only accessed through separate download), and our inclusion of them in the main body is consistent with standard APA style reporting. Furthermore, we do not agree that these data are not important to the research goal, and instead suggest that these data are important for informing the reader as to the nature of the sample, and the relationships between the main study variables. The inclusion of primary demographic and main variable data in the main body of the manuscript is also consistent with our own publications, and those of other studies published in PLoS One. We are therefore very hesitant and prefer not to remove these tables from the main manuscript. 

4. On page 6, there is a sentence that says “ASD traits were found to significantly correlate with suicidal behaviour, and the relationship was mediated by burdensomeness and thwarted belonging….. ” Does it mean burdensomeness and thwarted belonging are two of the underlying mechanisms explaining the relationship between autism and suicidal thoughts? If yes, then why this is not addressed in the next section of the paper, which is about “mechanisms underpinning suicide risk and the autistic phenotype”?

Response: We agree with the Reviewer that the referenced study could have potentially been described in the section “mechanisms underpinning suicide risk and the autistic phenotype” as the study addresses a mechanism from ASD traits to suicidality based on Joiner’s (2005) theory. However, we pose that this study (Pelton & Cassidy, 2017) is better placed under the subheading “Autism phenotype as a risk factor for suicide”. First, the study examined the path from autistic traits through burdensomeness/belonging to suicidality, and therefore fits equally well within the section on the contribution of autistic traits to suicidality. Second, the next section, on mechanisms, builds an argument for the role of the autistic phenotype i.e., from those constructs associated with the diagnostic criteria of the disorder to suicidality directly. Neither belonginess nor burdensomeness fit this criteria. As examined by Pelton and Cassidy, the path from ASD traits to burdensomeness/belonging to suicidality is an indirect mediation pathway. The factors we have identified are those dimensional constructs that we think may directly predict suicidal risk. While a more complex model might be constructed that includes mediation or other pathways though factors such as belonginess/burdensomeness etc., this was beyond the scope of our study. Therefore, we believe that this section (i.e., mechanisms) is better focused on the specific constructs we test in the present study.

5. There is a typo on page 8 under the “Participants” section. It says “…. condiucted recruitment online using…. ” “Condiucted” is a typo.

Response: Thank you for pointing this out, now corrected.

6. On page 15, please provide a citation for Little’s MCAR test. Also on page 15, I am confused why the authors found out that the data are not MCAR but still used listwise deletion.

Response: The reference for MCAR is now provided. We note that there was an error in our reporting of MCAR – the non-significant p-value (p = .895) does indeed indicate that data are missing completely at random. This has now been corrected in text. We removed cases with missing data based on the recommendations by Tabachnick & Fidell. Given that data were missing at random, we have not altered this methodology and apologise for the error. The sentence now reads:

“Little’s MCAR test indicated that data were missing completely at random, p = .895 [70].”

7. On page 8, the sentence, “The aims of the present study were to examine (1) the contribution of two key clinical domains––social communication difficulties, insistence on sameness––usually considered core features of ASD, but also present across a range of other disorders, that have been associated with suicidal risk and behavior, thereby deconstructing the impact of specific ASD domains on suicide risk (assessed using DSM-5 suicidal ideation; SI)….”, is very confusing to read. Please simplify the sentence.

Response: We have significantly revised this sentence for clarity. The paragraph now reads:

“The aims of the present study were to examine (1) the contribution of the two core clinical domains of ASD—social communication difficulties, insistence on sameness—on suicide risk (assessed using DSM-5 suicidal ideation; SI) and (2) the additional contribution and interaction of two key dimensional constructs––cognitive control and rumination. We predict that each of the identified constructs will independently contribute to SI, controlling for depression.”

---

## [Decision Letter · Decision Letter 1]

30 Nov 2020

PONE-D-20-29676R1

Domains of the autism phenotype, cognitive control, and rumination as transdiagnostic predictors of DSM-5 suicide risk

PLOS ONE

Dear Dr. Hedley,

Thank you for submitting your manuscript to PLOS ONE. After careful consideration, we feel that it has merit but does not fully meet PLOS ONE’s publication criteria as it currently stands. Therefore, we invite you to submit a revised version of the manuscript that addresses the points raised during the review process.

We look forward to receiving your revised manuscript.

Kind regards,

Vincenzo De Luca

Academic Editor

PLOS ONE

Reviewers' comments:

Reviewer's Responses to Questions

**Comments to the Author**

1. If the authors have adequately addressed your comments raised in a previous round of review and you feel that this manuscript is now acceptable for publication, you may indicate that here to bypass the “Comments to the Author” section, enter your conflict of interest statement in the “Confidential to Editor” section, and submit your "Accept" recommendation.

Reviewer #1: (No Response)

2. Is the manuscript technically sound, and do the data support the conclusions?

Reviewer #1: Partly

3. Has the statistical analysis been performed appropriately and rigorously? 

Reviewer #1: No

4. Have the authors made all data underlying the findings in their manuscript fully available?

Reviewer #1: Yes

5. Is the manuscript presented in an intelligible fashion and written in standard English?

Reviewer #1: Yes

6. Review Comments to the Author

Reviewer #1: The manuscript has improved. However, there are still a few things that need to be addressed.

First, on page 15, it says that "Little's MCAR test indicated that data were missing completely at random." Actually, non-significant test does NOT indicate the data are MCAR although it is a common misconception. A non-significance result only means that the covariance structure of the data across different missing data patterns are not heterogenous. Please refer to Yuan, Jamshidian and Kano (2018)'s paper about this. In the next revision, please correct this.

Second, please mention in your results section how to access your data (i.e., provide the "figshare" link).

Third, on page 20, I am not sure what those "t" values are. Please clarify.

Finally, on page 20, please also elaborate more on your results. Interpret the regression coefficients, and explain what each coefficient means based on your research purposes.

7. PLOS authors have the option to publish the peer review history of their article (what does this mean?). If published, this will include your full peer review and any attached files.

Reviewer #1: No

---

## [Author Response · Author response to Decision Letter 1]

30 Nov 2020

Dear Editor,

Thank you for the opportunity to revise and re-submit our manuscript. Please find below a detailed response to all comments. We thank the reviewer for the additional suggestions and have revised the text accordingly. In particular, we have significantly elaborated on the interpretation of the regression analysis, as well as attending to the other three minor suggestions.

Response to Reviewer

1. First, on page 15, it says that "Little's MCAR test indicated that data were missing completely at random." Actually, non-significant test does NOT indicate the data are MCAR although it is a common misconception. A non-significance result only means that the covariance structure of the data across different missing data patterns are not heterogenous. Please refer to Yuan, Jamshidian and Kano (2018)'s paper about this. In the next revision, please correct this.

Response: The text has been revised.

Revised text: “No more than 1% (M = .303, SD = .19, Range = 0–1%) of data were missing for any questionnaire item overall, and Little’s MCAR test was not significant, p = .895 [70]. Thus, following Tabachnick and Fidell [71], cases with missing data on any of the questionnaires were deleted (n = 77, 3.8%).”

2. Second, please mention in your results section how to access your data (i.e., provide the "figshare" link).

Response: added.

Inserted text: “The data that support the findings of the study are openly available in “Figshare” at https://doi.org/10.26181/5e992fc659d7c [90].”

3. Third, on page 20, I am not sure what those "t" values are. Please clarify.

Author response: In response to the reviewer’s comment, t-statistics from the regression analysis have been deleted.

4. Finally, on page 20, please also elaborate more on your results. Interpret the regression coefficients, and explain what each coefficient means based on your research purposes.

Response: Results on p20 have been significantly elaborated upon.

Revised text: “Table 4 presents the results of the linear regression model predicting SI. All hypothesized constructs were included in the model. Age was controlled for by including it in the model. The full model accounted for 43.3% of variance in SI scores, F(7, 1843) = 201.19, p < .001. Social Communication Difficulties significantly predicted SI, with the b-weight revealing that for each unit increase in Social Communication Difficulties, SI increased by 0.085 units. Similarly, Insistence on Sameness was also identified as a significant predictor of SI, with the b-weight revealing that for each unit increase in Insistence on Sameness, SI increased by 0.012 units. Attentional and Inhibitory Control both significantly predicted SI, with the b-weights revealing that for each unit increase in Attentional Control, SI decreased by 0.015 units, and for each unit increase in Inhibitory Control, SI increased by 0.015 units. Rumination was not a significant predictor of SI when entered in the model with the other variables, with each unit increase in Rumination associated with a decrease in SI of -0.010 units. Overall, Depression made the largest contribution to SI (beta = 0.484). Comparing Social Communication to Insistence on Sameness; Social Communication Difficulties (beta = 0.192) was relatively more important than Insistence on Sameness (beta = 0.144). These two core variables shared some variance, but correlations in Table 3 reveal that these were largely independent contributions. Attentional Control (beta = -0.085) and Inhibitory Control (beta = 0.077) made similar, yet relatively smaller contributions to the model.”

---

## [Decision Letter · Decision Letter 2]

21 Dec 2020

PONE-D-20-29676R2

Domains of the autism phenotype, cognitive control, and rumination as transdiagnostic predictors of DSM-5 suicide risk

PLOS ONE

Dear Dr. Hedley,

Thank you for submitting your manuscript to PLOS ONE. After careful consideration, we feel that it has merit but does not fully meet PLOS ONE’s publication criteria as it currently stands. Therefore, we invite you to submit a revised version of the manuscript that addresses the points raised during the review process.

We look forward to receiving your revised manuscript.

Kind regards,

Vincenzo De Luca

Academic Editor

PLOS ONE

Reviewers' comments:

Reviewer's Responses to Questions

**Comments to the Author**

1. If the authors have adequately addressed your comments raised in a previous round of review and you feel that this manuscript is now acceptable for publication, you may indicate that here to bypass the “Comments to the Author” section, enter your conflict of interest statement in the “Confidential to Editor” section, and submit your "Accept" recommendation.

Reviewer #1: (No Response)

2. Is the manuscript technically sound, and do the data support the conclusions?

Reviewer #1: Yes

3. Has the statistical analysis been performed appropriately and rigorously? 

Reviewer #1: Yes

4. Have the authors made all data underlying the findings in their manuscript fully available?

Reviewer #1: (No Response)

5. Is the manuscript presented in an intelligible fashion and written in standard English?

Reviewer #1: Yes

6. Review Comments to the Author

Reviewer #1: First, when I clicked on the link for the data (https://doi.org/10.26181/5e992fc659d7c [90].), it led me to a page with an error message. Please upload your data on the Open Science Framework (OSF) and provide the appropriate link.

Second, please use correct APA notations for mathematical symbols. Don't just write "beta=0.144".

7. PLOS authors have the option to publish the peer review history of their article (what does this mean?). If published, this will include your full peer review and any attached files.

Reviewer #1: No

---

## [Author Response · Author response to Decision Letter 2]

21 Dec 2020

Dear Editor,

Thank you for the opportunity to revise and re-submit our manuscript. Please find below a detailed response to all comments. 

Response to Reviewer

1. First, when I clicked on the link for the data (https://doi.org/10.26181/5e992fc659d7c [90].), it led me to a page with an error message. Please upload your data on the Open Science Framework (OSF) and provide the appropriate link.

Response: It is unclear why the Reviewer was unable to access these data, which were hosted on the University OP[A]L Open framework. We wonder whether the inclusion of the reference number [90] in the link caused the error message? Alternatively, it is possible that the Reviewer was unable to open the file as it was in SPSS? Nonetheless, at the Reviewer’s request we have transferred the file to OSF and have replaced the OP[A]L link with the new OSF link. We have additionally saved the file in .spss, .xlsx and .csv formats, and updated the referencing and links. We have additionally double checked and confirmed the links are working and the files are readily downloadable from OSF. The new link is: https://doi.org/10.17605/OSF.IO/C2AP3

2. Second, please use correct APA notations for mathematical symbols. Don't just write "beta=0.144".

Response: Replaced.

---

## [Decision Letter · Decision Letter 3]

4 Jan 2021

Domains of the autism phenotype, cognitive control, and rumination as transdiagnostic predictors of DSM-5 suicide risk

PONE-D-20-29676R3

Dear Dr. Hedley,

We’re pleased to inform you that your manuscript has been judged scientifically suitable for publication and will be formally accepted for publication once it meets all outstanding technical requirements.

Kind regards,

Vincenzo De Luca

Academic Editor

PLOS ONE

Additional Editor Comments (optional):

Reviewers' comments:

Reviewer's Responses to Questions

**Comments to the Author**

1. If the authors have adequately addressed your comments raised in a previous round of review and you feel that this manuscript is now acceptable for publication, you may indicate that here to bypass the “Comments to the Author” section, enter your conflict of interest statement in the “Confidential to Editor” section, and submit your "Accept" recommendation.

Reviewer #1: All comments have been addressed

2. Is the manuscript technically sound, and do the data support the conclusions?

Reviewer #1: Yes

3. Has the statistical analysis been performed appropriately and rigorously? 

Reviewer #1: Yes

4. Have the authors made all data underlying the findings in their manuscript fully available?

Reviewer #1: Yes

5. Is the manuscript presented in an intelligible fashion and written in standard English?

Reviewer #1: Yes

6. Review Comments to the Author

Reviewer #1: (No Response)

7. PLOS authors have the option to publish the peer review history of their article (what does this mean?). If published, this will include your full peer review and any attached files.

Reviewer #1: No

---

## [Editor Report · Acceptance letter]

6 Jan 2021

PONE-D-20-29676R3 

Domains of the autism phenotype, cognitive control, and rumination as transdiagnostic predictors of DSM-5 suicide risk 

Dear Dr. Hedley:

I'm pleased to inform you that your manuscript has been deemed suitable for publication in PLOS ONE. Congratulations! Your manuscript is now with our production department. 

Kind regards, 

on behalf of

Dr. Vincenzo De Luca 

Academic Editor

PLOS ONE